# Molecular Biomarkers in Multiple Sclerosis and Its Related Disorders: A Critical Review

**DOI:** 10.3390/ijms21176020

**Published:** 2020-08-21

**Authors:** Maryam Gul, Amirhossein Azari Jafari, Muffaqam Shah, Seyyedmohammadsadeq Mirmoeeni, Safee Ullah Haider, Sadia Moinuddin, Ammar Chaudhry

**Affiliations:** 1Precision Rheumatology INC, 2050 South Euclid Street, Anaheim, CA 92802, USA; 2Student Research Committee, School of Medicine, Shahroud University of Medical Sciences, Shahroud 3614773947, Iran; ah.azari.7592@gmail.com (A.A.J.); sadeqmirmoeeni@gmail.com (S.M.); 3Deccan College of Medical Sciences, P.O. Kanchanbagh, DMRL ‘X’ Road, Santhosh Nagar, Hyderabad 500058, Telangana State, India; muffaqamshah@gmail.com; 4Shaikh Khalifa Bin Zayed Al-Nahyan Medical College, Shaikh Zayed Medical Complex, Lahore 54000, Pakistan; safeeullahhaider@gmail.com; 5Department of Internal Medicine, San Antonio Regional Medical Center, 999 San Bernardino Rd, Upland, CA 91786, USA; Sadia.moinuddin@gmail.com; 6Department of Radiology, City of Hope National Medical Center, 1500 East Duarte Road, Duarte, CA 91010, USA; achauhdry@coh.org

**Keywords:** multiple sclerosis, MS, biomarker, molecular, pathophysiology, prognosis, treatment, mimics, differential diagnosis

## Abstract

Multiple sclerosis (MS) is a chronic autoimmune disease affecting the central nervous system (CNS) which can lead to severe disability. Several diseases can mimic the clinical manifestations of MS. This can often lead to a prolonged period that involves numerous tests and investigations before a definitive diagnosis is reached. As well as the possibility of misdiagnosis. Molecular biomarkers can play a unique role in this regard. Molecular biomarkers offer a unique view into the CNS disorders. They help us understand the pathophysiology of disease as well as guiding our diagnostic, therapeutic, and prognostic approaches in CNS disorders. This review highlights the most prominent molecular biomarkers found in the literature with respect to MS and its related disorders. Based on numerous recent clinical and experimental studies, we demonstrate that several molecular biomarkers could very well aid us in differentiating MS from its related disorders. The implications of this work will hopefully serve clinicians and researchers alike, who regularly deal with MS and its related disorders.

## 1. Multiple Sclerosis (MS)

Multiple sclerosis (MS) is a chronic autoimmune disease affecting the central nervous system (CNS), caused by genetic and environmental factors. It is characterized by intermittent and recurrent episodes of inflammation that result in the demyelination and subsequent damage of the underlying axons present in the brain, optic nerve and spinal cord [1,2,3]. There are four courses of MS: (a) relapsing–remitting, (b) primary progressive, (c) secondary progressive and (d) progressive relapsing [3]. There are a number of illnesses that can mimic the clinical manifestations of MS and this can include several pathologic processes because they all share a common pathway of demyelination and ensuing damage to the underlying axons present in the brain, optic nerve and spinal cord [1,2,3,4]. Over the years, the similarity of symptoms has led to numerous instances of neurologic conditions being misdiagnosed as MS and vice versa [1,3]. The McDonalds criteria established in 2001 aimed to rectify this by setting guidelines for MS diagnosis [5]. However, because MS is often a diagnosis of exclusion, there is still a considerable portion of time where patients may take unnecessary medications for other plausible diseases before a diagnosis is established [1,3,6,7]. With the advent of immunomodulating therapy, it has become more important to diagnose or even exclude MS more effectively earlier on in the course of the illness [1,3,6,7]. By being aware and cognizant of the various diseases which mimic MS, this can empower clinicians and researchers to help deliver accurate counseling and treatment to their patients for their specific diagnosis [8].

### 1.1. Biomarkers in Pathophysiology of MS

One of the most prominent biomarkers of MS are miRNAs, which target several protective or pathogenic signaling pathways, and they have been found to be upregulated or downregulated in MS patients [9,10]. It is reported that protective miR-199a, pathogenic miR-320, miR-155, miR-142-3p and miR-142 are increased in MS lesions or peripheral blood mononuclear cells (PBMCs) [9,10,11]. On the other hand, miR-219, miR-34a, miR-103, miR-182-5p, miR-124 and miR-15a/b are decreased in the Cerebrospinal fluid (CSF), Tregs or PBMCs of MS patients [9,10,12,13,14]. There is a strong correlation between miRNAs and various manifestations of MS such as cognitive dysfunction and oxidative status, which can result in depression or fatigue [9,10,11,12,13,14].

Another oxidative biomarker is nicotinamide adenine dinucleotide phosphate (NADPH) oxidase 2 (Nox2); an enzyme that catalyzes the reduction of oxygen to produce reactive oxygen species, which plays a role in the pathogenesis of MS [15].

Recent research has suggested that reactive T cells directed against neuronal protein β-synuclein can invade and destroy the gray matter, which is a hallmark of MS [16]. In addition, a high level of β-synuclein reactive T cells in the peripheral blood of MS patients indicates that this biomarker has a key role in provoking T cells in MS [16].

### 1.2. Biomarkers in Diagnosis of MS

It has been suggested that the levels of specific complement proteins such as C1q, C3d and C5b-9 in the serum and CSF could potentially serve as novel biomarkers to diagnose the various MS subtypes and determine the disease activity [17]. miRNAs levels such as low miR-219 and high miR-150 in CSF are novel biomarkers that can distinguish MS from other neurologic conditions [18,19].

Recent studies have also shown that brain-derived neurotrophic factor (BDNF) and soluble isoform of the interferon-β (IFN-β) receptor (sIFNAR2) levels may serve as useful biomarkers for the diagnosis of MS [20,21].

### 1.3. Biomarkers in Treatment and Prognosis of MS

Several studies demonstrate that a vast number of immune modulators or oxidative stress biomarkers can be used as therapeutic targets and for further studies on MS. The most notable examples are miR-497-5p, semaphorin-3A, coenzyme Q10, interferon gamma-stimulated dendritic cell exosomes (IFNγ-DC-Exos), glutathione (GSH) and dimethyl fumarate (DMF) [22,23,24,25,26].

It has been suggested that GSH (the major antioxidant in the brain) can be used for therapeutic applications as well as to predict and monitor the disease progression [26]. Monitoring disease activity in MS can be done by the use of total antioxidant status (TAS), high levels of total hydroperoxides and ceruloplasmin transferrin ratio (Cp:Tf) ratio (strictly related to Fe management) [27,28].

The potential biomarkers which can be used to predict the prognosis of relapsed or progressive forms of MS, as well as the responsiveness to treatment in patients with MS are SIRT1 (a NAD-dependent deacetylase sirtuin-1) mRNA, *Response gene to complement-32 (RGC-32)*, Fasl, IL-21, Tau proteins (proteins that stabilize microtubules), miR-191-5p, miR-128-3p and serum netrin-1 (an axon guidance protein) [17,21,29,30,31,32].

The primary CSF biomarkers which can be used to predict the prognosis of MS are β-amyloid (Aβ) levels, neurofilament light (NF-L), neurofilament heavy (NF-H), chitinase 3-like-1 (CHI3L1) and immunoglobulin M (IgM) [33,34]. Elevated levels of lysophosphatidic acid (LPA) in the serum and CSF of relapsed MS patients can also be used as biomarkers to monitor the disease activity [35].

### 1.4. Systemic Lupus Erythematosus: Neuropsychiatric SLE

The nervous system is one of the major organs affected in patients with systemic lupus erythematosus (SLE) [36]. Neuropsychiatric SLE (NPSLE) is a major cause of morbidity and mortality in patients with SLE and it can be hard to distinguish NPSLE from MS [36].

It is proposed that many autoimmune biomarkers like anti-NR2A antibodies, anti-dsDNA antibody, SLEDAI, complement levels, anti-ribosomal P antibodies (anti-P) and possibly anti-endothelial cell antibodies (ECA) have a potential role in the NPSLE pathogenesis [37,38].

A recent study showed that serum brain-derived neurotrophic factor (BDNF) level is a useful biomarker not only to diagnose NPSLE from other neurological diseases like MS, but also to monitor the prognosis and response to treatment [39]. In addition, it was found that anti-U1 ribonucleoprotein (RNP) antibodies (anti-U1RNP Abs) in the CSF could be a useful diagnostic tool for central NPSLE [40].

It is determined that serum anti-dsDNA antibodies can predict the development and prognosis of NPSLE [41]. Recently, the small colony-stimulating factor-1 receptor (CSF-1R) kinase inhibitor has been used in experiments on lupus-prone mice. Results showed that it could reduce the NPSLE manifestations such as depression-like behavioral deficits [42].

### 1.5. Antiphospholipid Syndrome

Antiphospholipid syndrome (APS, or Hughes syndrome) is a systemic autoantibody-mediated acquired thrombophilia that is characterized by the presence of antiphospholipid antibodies (APL) against phospholipid-binding plasma proteins, such as beta-2-glycoprotein I (b2GPI) [43,44,45,46]. Because of its neurological symptoms which include strokes, migraine, memory loss and the common clinical, laboratory and radiological features; it has the potential to be misdiagnosed with other neurological diseases especially MS [44,45,46,47].

In the pathophysiology aspect, the biomarker b2GPI has been identified as the major autoantigen of APS, but there are other antibodies that bind to phosphatidylserine, prothrombin and thrombomodulin [45,46,47,48]. The laboratory biomarker tests for the diagnosis and management of APS are anti-beta-2-glycoprotein I (anti-b2GPI), anti-cardiolipin (ACL) and lupus anticoagulant (LAC) [45,46,48,49].

### 1.6. Neuro-Behçet’s Disease

Neuro-Behçet’s disease (NBD) is not very common in the course of Behçet’s disease (BD, a systemic variable vessel vasculitis), however, it is associated with significant mortality and morbidity [50].

Several biomarkers like interferon-alpha, IL-6 receptor blocking agent (tocilizumab), and anti-TNF agents are the most preferred options for treatment [50,51]. In a retrospective study, most patients with NBD exhibited *Mediterranean fever (MEFV)* gene mutations which can act as a disease modifier in NBD and it can be targeted with CNS anti-inflammatory treatments especially colchicine [52]. *Defensin alpha 1B (DEFA1B)* and *NOD-like receptor protein 3 (NLRP3)* gene expressions increase during NBD attacks so they can be used as biomarkers to predict disease activity and also as possible targets for future therapeutic experiments [53].

### 1.7. Primary Angiitis of the CNS

Primary angiitis of the CNS (PACNS) is an extremely rare disorder resulting in inflammation and destruction of CNS vessels without any evidence of vasculitis outside the CNS and there is no valuable biomarker test to diagnose PACNS [54,55].

Recently, the efficacy and safety of a new biomarker mycophenolate mofetil (MMF) for achieving and maintaining remission of childhood PACNS have been demonstrated [56]. Furthermore, high levels of a protein biomarker neurofilament light chain (NFL) in the sera (CSF) indicates neuroaxonal damage, and its predictive value has been shown recently [57].

### 1.8. Polyarteritis Nodosa

Polyarteritis nodosa (PAN), is a necrotizing vasculitis mainly manifested by peripheral neuropathy (most frequent and earliest), weight loss, fever, cutaneous lesions, hypertension and cardiac failure [58].

One of the suggested biomarkers in PAN is matrix metalloproteinase-3 (MMP-3) which can be used in the diagnosis, management and following up on the PAN prognosis [59]. A recent study demonstrated that high serum lysosomal-associated membrane protein-2 (LAMP-2) could be a useful biomarker for evaluating the prognosis and the disease activity of PAN [60].

### 1.9. Antineutrophil Cytoplasmic Antibodies (ANCA)-Associated CNS Vasculitis (AAV)

Anti-neutrophil cytoplasmic antibody (ANCA) associated vasculitis (AAV), is a systemic small-vessel vasculitis by pathogenic ANCA production, consists of granulomatosis with polyangiitis (GPA), microscopic polyangiitis (MPA) and eosinophilic granulomatosis with polyangiitis (EGPA) and Neurologic involvement [61,62].

The ANCA is the main pathogenic biomarker of AAV which targets proteinase 3 (PR3) and myeloperoxidase (MPO) in the cytoplasm of neutrophils and monocytes [63]. ANCA is also a strong biomarker for diagnosis and even the monitoring of AVV activity [64].

The treatment goal of AVV is to inhibit the ANCA so it mainly consists of high-dose corticosteroids, cyclophosphamide or rituximab [65].

### 1.10. Acute Disseminated Encephalomyelitis (ADEM)

Acute disseminated encephalomyelitis (ADEM) is a monophasic demyelinating disorder of the CNS with a high incidence in children [66,67,68].

The elevated cytokines and chemokines (involved in helper T cells, regulatory T cells and B cell pathways) in CSF suggest the possible role of these biomarkers on the pathogenesis of the ADEM [66,68].

Until recently there was no diagnostic biomarker test to confirm ADEM [68,69]. Moreover, there are no clinical features or biomarkers that can reliably differentiate ADEM from MS at the first attack [67,70]. In spite of this, MS can occur after ADEM in up to 17% of cases, which indicates the importance of differentiating ADEM from MS early on [66]. Several studies have shown that some new biomarkers like myelin oligodendrocyte glycoprotein (MOG) antibody, serum IgG targeting myelin basic protein (MBP) and myelin-associated oligodendrocyte basic protein (MOBP) can be used to diagnose and differentiate ADEM from MS at the first attack [67,69,71,72].

### 1.11. Neuromyelitis Optica (NMO)

Neuromyelitis Optica (NMO, Devic’s disease) is an idiopathic autoimmune disorder of the CNS presenting with optic neuritis, transverse myelitis and possibly medullary involvement and narcolepsy [73,74].

The pathogenesis of NMO can be attributed to the anti-aquaporin 4 (anti-AQP4) antibodies that cross the blood–brain barrier and damage the astrocyte foot processes. IL-6 may also play a role [73]. IL-6 is produced by the macrophages and dendritic cells and promotes differentiation of T cells into Th-17 cells. The IL-6 levels in NMO have been found to be higher than those in optic neuritis and related disorders suggesting a greater role in disease pathogenesis [75].

Immunosuppressants are used for treatment. High dose steroids may be used acutely. In refractory cases, exchange plasmapheresis [76] and intravenous or subcutaneous tocilizumab, an interleukin-6 antagonist has shown promise [77,78].

### 1.12. Optic Neuritis

Optic neuritis is a disease most commonly seen in women presenting as pain with extraocular movements and potentially progressing to partial or complete loss of vision [79]. Optic neuritis usually resolves within 3–6-week on its own however it may also progress into MS [80].

The pathology is inflammatory damage to the optic nerve and is confirmed by the presence of anti-myelin oligodendrocyte glycoprotein (anti-MOG) or anti-AQP4 antibodies. It may also be a manifestation of systemic lupus erythematosus, sarcoidosis or immune-mediated vasculitis [81,82].

Typical optic neuritis is usually self-limited and resolves on its own without any clinical treatment. The use of steroids may reduce the duration of the disease [83]. However, it is debatable and the treatment plan should be based on the biomarker [84].

### 1.13. Vitamin B12 Deficiency

Vitamin B12 (cobalamin) deficiency presents with a range of signs from megaloblastic anemia and macrocytic anemia to subacute combined degeneration of the spinal cord [85].

If diagnosed early the prognosis is good, however, delayed treatment may lead to permanent neurological dysfunction [86].

Elevated serum methylmalonic acid and homocysteine levels lead to neuronal damage and can be detected as biomarkers [87,88]. Holotranscobalamin is used as a biomarker since B12 levels alone may be misleading [89]. Anti-intrinsic factor antibodies (AIFA) and anti-parietal cell antibodies (APCA) may also be useful when pernicious anemia is the cause [85]. Intramuscular injections and oral replacement are used to treat deficiency [90].

### 1.14. Cerebral Autosomal Dominant Arteriopathy with Subcortical Infarcts and Leukoencephalopathy (CADASIL)

Cerebral autosomal dominant arteriopathy with subcortical infarcts and leukoencephalopathy (CADASIL) is a rare autosomal dominant (AD) disease caused by mutations in the *NOTCH3* gene of chromosome (Chr.) 19p13.2-p13.1 [91]. It affects small vessels of the vascular system causing diffuse microangiopathy [92,93].

Biomarkers that have shown promising results are the Von Willebrand factor (vWF), endothelial progenitor cells (EPCs) and circulating progenitor cells (CPCs). The levels of vWF levels were significantly higher, while EPCs and CPCs levels were reduced [92].

Management is solely pragmatic and no therapies have been able to limit the disease progression. The biomarkers are not suitable prognostic indicators. Increases in the number of ischemic attacks worsen the prognosis [94].

### 1.15. Leber’s Hereditary Optic Neuropathy (LHON)

Leber’s hereditary optic neuropathy (LHON) is a rare mitochondrial disease characterized by progressive, bilateral and typically painless central vision loss [95,96].

Some promising/experimental biomarkers are low arylsulfatase A levels and elevated CSF/urine sulfatide levels. These biomarkers are also present in other diseases, hence more research is warranted in the search for a specific biomarker. Nonetheless, these biomarkers are still useful when formulating a focused differential diagnosis when presented with the varying signs of a neurodegenerative disease. [95,97,98].

The current standard of care is therapies with idebenone/quinine analogs, 4-aminopyridine (4-AP), gene therapy and stem cell therapy [99,100].

The prognosis is generally poor, and most patients will end up legally blind with visual acuity of 20/200 or worse. Biomarker levels do not correlate well with the disease progression [101,102].

### 1.16. Mitochondrial Encephalomyopathy with Lactic Acidosis and Stroke-like Episodes (MELAS)

Mitochondrial encephalomyopathy with lactic acidosis and stroke-like episodes (MELAS) is a multiorgan disease with stroke-like episodes, dementia, epilepsy, lactic acidemia, myopathy and recurrent headaches [103]. The MT-TL1 mutation accounts for most cases [104]

Broad yet unspecific biomarkers mentioned are [104,105,106,107,108]: Varying levels of serum lactate, CSF lactate/pyruvate ratio, Ventricular lactic acid (LA), creatine kinase (CK), nitric oxide (NO), N-acetyl-L-aspartate (NAA) and total choline (*t*-Cho). The most novel one is serum miR-27b-3p.

There is no known cure or consensus on the proper approach for treating MELAS. L-arginine is the most commonly used/effective agent. Some benefits are noted with CoQ, idebenone, vitamins B/C/E and levocarnitine. The prognosis is highly variable, and the biomarker levels do not correlate well with disease progression [103,104,109,110].

### 1.17. Metachromatic Leukodystrophy (MLD)

Metachromatic leukodystrophy (MLD)is an autosomal recessive (AR) lysosomal storage disease. It is caused by mutations in the ARSA region on Chr. 22q13.33. This causes a deficiency of the arylsulfatase-A (ASA) enzyme and the accumulation of sulfatides, leading to demyelination [111].

The most reliable/mentioned biomarkers are low arylsulfatase A levels and elevated CSF/urine sulfatide levels. These are mentioned numerous times in the literature and warrant a strong suspicion for MLD [112,113,114,115].

There is no treatment available for MLD. Hematopoietic stem cell transplantation (HSCT), gene therapy and enzyme replacement therapy (ERT) are the options currently in the trial phase [116,117,118]. Biomarker levels do not correlate well with the disease progression and cannot serve as prognostic indicators [119].

### 1.18. Krabbe’s Leukoencephalopathy (KL)

Krabbe’s leukoencephalopathy (KL) is a rare AR disorder with widespread demyelination. Mutations on Chr. 14q31 cause a deficiency of the β-galactosylcerebrosidase (GALC) enzyme [115,120,121].

There are elevated blood levels of galactosphingosine (psychosine) and reduced GALC enzyme activity. Elevated psychosine levels correlate well with disease severity (possible prognostic indicator). It is specific to confirm the diagnosis, along with clinical manifestations and imaging [120,122,123,124,125].

Currently, the only approved treatment for presymptomatic infantile and late-onset patients is HSCT. The prognosis varies based on how early the diagnosis is made, how quickly HSCT was started (10%–20% mortality rate associated with HSCT) and the severity of the disease progression [115,125].

### 1.19. Multiple Sulfatase Deficiency (MSD)

Multiple sulfatase deficiency (MSD) is a rare inherited metabolic disorder caused by a deficiency in the formylglycine-generating enzyme (FGE), due to mutations in the *sulfatase modifying factor 1 (SUMF1)* gene on Chr. 3p26.1. Leading to a deficiency in all molecules of the sulfatase enzyme family. This results in increased sulfated lipids and mucopolysaccharides buildup in various cells leading to dysfunction [126,127].

The major trends of biomarker levels mentioned are [126,127,128,129,130,131,132,133,134,135,136]: (a) elevation of sulfatides in the urine; (b) deficiency of sulfatase enzyme activity in the leukocytes; (c) elevation of glycosaminoglycan (GAG) in the urine and at times plasma (variable).

There is no specific therapy available for MSD, but ERT and gene therapy options are being explored [127,136]. Prognostic indicators are based on FGE stability and residual enzyme activity. Those with drastic impairments in both had a worse prognosis, whereas those with residual activity/levels had a better prognosis.

### 1.20. Alexander Disease (AxD)

AxD is a rare, progressive and generally fatal neurological disorder. Resulting from dominant mutations affecting the coding region of GFAP, which encodes for glial fibrillary acidic protein (GFAP). GFAP is the major intermediate filament protein of astrocytes in the CNS [137].

The biomarker most mentioned was GFAP. The levels were consistently elevated in the CSF, but only occasionally elevated in the blood [137,138,139,140,141].

Management revolves around supportive care, dealing with various CNS dysfunctions and an emphasis on genetic counseling [138,141]. Experimental work is being done on using GFAP as a target for pharmacological agents [141]. GFAP levels do not serve as an accurate prognostic indicator, because elevated levels do not correlate well with disease severity.

### 1.21. Adrenoleukodystrophy (ALD)

Adrenoleukodystrophy (ALD) is an X-linked disorder caused by mutations in the *ABCD1* gene which encodes the adrenoleukodystrophy protein (ALDP). A defect in ALDP results in elevated levels of very-long-chain fatty acids (VLCFAs) and their accumulation in plasma and tissues [142,143].

The biomarkers mentioned are elevated levels of plasma and tissue VLCFAs and elevated levels of C26:0 lysophosphatidylcholine (C26:0-lyso-PC) [144,145,146,147]. Huffnagel et al. found that C26:0-lysoPC is a better and more accurate biomarker than plasma VLCFA levels [145].

There is no curative treatment available for ALD. HSCT remains the only therapeutic intervention. There is no treatment for any associated progressive myelopathy. Early diagnosis of males with ALD by screening at birth allows for the early detection of adrenal insufficiency [144,145].

The levels of the biomarkers mentioned cannot be used as prognostic indicators [144,145]. There is variability noted in the literature for disease progression [146,147,148].

### 1.22. Pelizaeus-Merzbacher Disease (PMD)

Pelizaeus-Merzbacher disease (PMD) is an X-linked disorder caused by mutations in the *proteolipid protein 1 (PLP1)* gene on Chr. Xq22, which encodes for PLP1. PLP1 plays an integral role in compact myelin sheath stabilization [149].

Mierzejewska et al. state that the diagnosis can be confirmed only with molecular analysis of the genetic mutation/abnormality [150]. A thorough search of the literature did not yield any reliable evidence of a diagnostic biomarker for PMD. However, some experimental results are being reported. Takanashi et al. found that on magnetic resonance spectroscopy (MRS), an LCModel revealed increased total N-acetyl-L-aspartate (tNAA), creatine (Cr), myo-inositol (min), choline (Cho) [151].

There are no effective therapies for PMD. Genetic counseling and supportive care for disabilities/special needs are the current protocols [149,152,153]. White-matter atrophy serves as a marker of clinical severity, and a prognostic indicator [152,153].

### 1.23. Chronic Lymphocytic Inflammation with Pontine Perivascular Enhancement Responsive to Steroids (CLIPPERS)

The immunopathogenesis of most inflammatory CNS disorders, like Chronic lymphocytic inflammation with pontine perivascular enhancement responsive to steroids (CLIPPERS) remains poorly understood [154]. Hence, there is not a useful diagnostic and prognostic biomarker test as of yet [154].

CLIPPERS is specified by determining punctate or curvilinear gadolinium-enhancing MRI lesions that are most noted in the pons [155,156].

Analysis of cerebrospinal fluid can determine mildly elevated protein, pleocytosis and oligoclonal bands or could be unremarkable. A CLIPPERS display can be connected with antibodies to MOG, but most cases are seronegative. Intravenous and oral steroids can improve clinical and paraclinical effects rapidly. Long-term steroid-sparing agents such as azathioprine and mycophenolate have been used as well, except for immunomodulatory monoclonal antibodies like tocilizumab and rituximab [155].

### 1.24. HIV-Related Disorders of the CNS

Chemokines and their receptors tend to play a key role for connection among neurons and inflammatory cells. Evidence is mounting for the connection between chemokines and their receptors in the progression of brain damage, including MS and HIV associated dementia. This system is therefore likely to become a serious goal for the treatment of neurodegenerative diseases. In the development of demyelinating lesions in MS patients, the RANTES polymorphism would contribute. In damaged tissues, RANTES is an inflammatory chemokine remarked to respond to pro-inflammatory cytokines or contact with pathogens. Its main role is the recruitment of effector cells to sites of inflammation. The data shows a strong association between the severity of dementia and CSF HIV viral load, suggesting the development of HIV-induced brain damage [157].

### 1.25. HTLV-1-Associated Myelopathy

The first human retrovirus that was associated with the development of cancer was the human T-cell lymphotropic virus (HTLV)-1, namely adult T-cell leukemia (ATL), the pathogenesis is still unknown [158].

HTLV-1-associated myelopathy/tropical spastic paraparesis (HAM/TSP) can be confirmed by HTLV-1 antibodies via western blot and/or a positive PCR. The finding of anti-HTLV-1 antibodies in CSF is necessary for the diagnosis of HAM/TSP. To rule out other diseases similar to HAM/TSP, like MS, neuroimaging is necessary but is not specific for HAM/TSP. Most treatments in HAM/TSP attempt to suppress or modulate the immune response or to reduce the HTLV-1 proviral load in this disorder. Some treatments have discovered that display improves in the progress of HAM/TSP, like corticosteroids, INF-α, mogamulizumab or monoclonal antibody against IL-2 Rα. Mogamulizumab is a synthesized anti-CCR4 monoclonal antibody, effectively decreased both the proviral load (PVL) and inflammatory activity in cells acquired from patients with HAM/TSP. Reduction in the PVL and spontaneous proliferation is caused by the Monoclonal antibody against IL-2 Rα. Given that HAM/TSP is characterized by activated T-cells in both the periphery and CNS, studies in HAM/TSP can give a chance for clarifying the pathogenesis of other neuroinflammatory disorders such as multiple sclerosis [159].

### 1.26. Lyme Borreliosis

Lyme borreliosis is a multisystem disease caused by spirochete *Borrelia burgdorferi* sensu lato [159]. Although *B. burgdorferi* can be one of the reasons for MS, it does not have a key role in the MS etiology [160]. Both Lyme borreliosis and MS can mimic lymphoma [160]. Rituximab can be useful for both MS and lymphoma, whether it could have a good effect on Lyme borreliosis or not [160].

Some investigation has shown that MR imaging results may be valuable for diagnostic workup of a patient with neuroborreliosis and T2 brain lesion indiscernible from those of MS [159].

### 1.27. Myelin Oligodendrocyte Glycoprotein IgG-associated Encephalomyelitis (MOG)

Myelin sheaths and oligodendrocytes express a protein that is named MOG. Anti-MOG antibodies (MOG-IgG) can cause demyelination in vitro and persuade experimental autoimmune encephalomyelitis. This disease is so rare and the incidence of that in children and adults is low [161].

The studies have shown that the brain lesion distribution criteria could be distinguishable in the MS patient from MOG-EM, even as early on as the disease onset [162].

### 1.28. Neurosyphilis

The CNS lesions in MS and syphilis disease are similar in histological appearance. Spirochetes can be the cause of both MS and syphilis, so the antibiotics could be useful for the treatment of these [162].

The most common characteristic of meningovascular syphilis is the atypical stroke-like presentation while the contrast-enhancing gammas in later stages may potentially look like MS. Both illnesses can result in CSF pleocytosis with elevated CSF gamma globulin and the presence of oligoclonal bands [163]. Even though the fact that neurosyphilis can still be the ‘‘great imitator’’, the presence or absence of a serum fluorescent treponemal antibody study (FTA-ABS) should provide essentially 100% sensitivity for infection with syphilis [163].

### 1.29. Progressive Multifocal Leukoencephalopathy

Progressive multifocal leukoencephalopathy (PML) caused by the JC virus (JCV) which is a fatal demyelinating disease of the CNS, this human polyomavirus can lytically infect and destroy the oligodendrocytes in immunosuppressed patients [164].

On the whole we can say that the RANTES polymorphism could be involved in the development of demyelinating lesions in NDLE pathology similarly to what was observed in MS patients, but not in PML [164].

## 2. Discussion

This review was written with the intended purpose of highlighting the current state of available evidence for the vast and growing sum of MS-related disorders. An emphasis was placed on featuring their associated biomarkers. As indicated in Table 1, we highlighted a wide array of molecular biomarkers that can aid us in the diagnosis, inform prognosis and act as guidelines for therapeutic management in MS and its related disorders. (Table 1)

Many of the mentioned biomarkers have strong evidence supporting their current use in the workup and management of their respective diseases. Despite the multitude of diseases that can potentially mimic MS, most patients express enough characteristic clinical evidence that in fact, there is only a limited need for an alternative diagnosis or supplementary testing/investigations.

The use of biomarkers in conjunction with the current standard of imaging modalities, namely MRI will hopefully allow for a quicker and more accurate diagnosis, as well as precise management strategies. The implications of this work will hopefully serve clinicians/researchers that are regularly dealing with MS and its related disorders. Further research is most definitely warranted for those diseases that have minimal to no biomarkers associated which can be relied upon confidently.

The overall data collated provides an accessory pathway that can be utilized in the management of MS and the diseases which mimic it. With continued research and work in this field, there is a possibility that biomarkers can become a routine and indispensable tool in the management of those diseases which historically did not rely upon such an avenue.

## Figures and Tables

**Table 1 ijms-21-06020-t001:** Summary of molecular biomarkers in multiple sclerosis and its related disorders (abbreviations are below table).

Type of Disease	Disease	Biomarkers				References
		Pathophysiologic	Diagnostic	Therapeutic	Prognostic	
**Autoimmune or inflammatory**	**MS**	miR-199a miR-320 miR-155 miR-142-3p miR-142 miR-219 miR-34a miR-103 miR-182-5p miR-124 miR-15a/b Nox2 β-synuclein	complement proteins:C1q, C3d and C5b-9 miR-219 miR-150 BDNF anti-U1RNP Abs sIFNAR2	miR-497-5p semaphorin-3A coenzyme Q10 IFNγ-DC-Exos GSH DMF	SIRT1 RGC-32 Fasl IL-21 Tau proteins miR-191-5p miR-128-3p serum netrin-1 β-amyloid NF-L NF-H CHI3L1 IgM LPA GSH TAS Cp:Tf	[9,10,11,12,14,15,16,17,18,19,20,21,22,23,24,25,26,27,28,29,30,31,32,35]
	**NPSLE**	anti-NR2A Ab anti-dsDNA Ab SLEDAI Complement levels anti-P Ab ECA	BDNF anti-U1RNP Abs	CSF-1R	BDNF anti-dsDNA Ab	[37,38,39,40,41]
	**APS**	APL Abs against b2GPI	anti-b2GPI ACL LAC	–	–	[43,44,45,46,48,49]
	**NBD**	MEFV gene mutations	–	IFN-α IL-6 receptor blocker anti-TNF agents MEFV gene mutations	DEFA1B gene NLRP3 gene	[50,51,52,53]
	**PACNS**	NF-L	–	MMF	–	[56,57]
	**PAN**	–	ESR MMP-3	ESR MMP-3	ESR MMP-3 LAMP-2	[59,60]
	**AAV**	ANCA	ANCA	inhibit the ANCA	ANCA	[61,62,63,64,65]
	**ADEM**	CSF elevated cytokines and chemokines	MOG Ab IgG targeting MBP MOBP	–	–	[66,67,68,69,71,72]
	**NMO**	anti-AQP4 IL-6	anti-AQP4	–	–	[73]
	**CLIPPERS**	Protein pleocytosis oligoclonal bands	-	–	–	[155]
	**MOG-EM**	anti-MOG Abs	-	-	-	[160]
**Infectious**	**HIV-related disorders of the CNS**	–	CSF HIV viral load	–	–	[157]
	**HAM/TSP**	HTLV-1 Abs	–	–	–	[158]
	**Neurosyphilis**	–	FTA-ABS	–	–	[163]
**Genetic disorders**	**CADASIL**	NOTCH3 gene mutation	vWF EPCs CPCs	–	–	[92]
	**LHON**	Arylsulfatase A CSF/urine sulfatide	Arylsulfatase A CSF/urine sulfatide	–	–	[95,97,98]
	**MELAS**	MT-TL1 mutation	Lactate CSF lactate/pyruvate ratio Ventricular LA CK NO NAA *t*-Cho	–	–	[104,105,106,107,108]
**demyelinating disorders**	**Optic neuritis**	anti-MOG	–	–	–	[81,82]
	**MLD**	Arylsulfatase A CSF/urine sulfatide	Arylsulfatase A CSF/urine sulfatide	–	–	[112,113,114,115]
	**KL**	Galactosphingosine (Psychosine)	Galactosphingosine (Psychosine)	–	Galactosphingosine (Psychosine)	[120,122,123,124,125]
	**MSD**	SUMF1 gene mutations FGE deficiency Sulfated lipids Mucopolysaccharides	Urine sulfatides Sulfatase enzyme deficiency Urine/Plasma GAG	–	FGE levels	[126,127,128,129,130,131,132,133,134,135,136]
	**AxD**	GFAP	GFAP	–	–	[137]
	**ALD**	VLCFAs	VLCFAs C26:0-lyso-PC	–	–	[142,143,144,145,146,147]
	**PMD**	PLP1 gene	tNAA Cr min Cho	–	–	[149,151]
**Other disorders**	**Vitamin B12 deficiency**	–	Methylmalonic acid Homocysteine Holotranscobalamin AIFA APCA	–	–	[85,87,88,89]

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
