# Peer review of "Molecular Biomarkers in Multiple Sclerosis and Its Related Disorders: A Critical Review"

_ijms, 2020, doi:10.3390/ijms21176020_

Round 1
Reviewer 1 Report
This review summarizes a handful of features of various diseases of CNS. Some of the disorders, by symptoms, overlap with MS, some not.
Comments
The authors chose an unconventional way to write this manuscript. Everything begging from language to definitions needs to be clarified.
The style of this work looks like a written out table. Each section needs a lot more information.
What are the definitions of different biomarkers?
There is a difference between proteins that play a role in the disease process and biomarker.
Words are abbreviated once in the text, and, once abbreviated, only abbreviations are used.
If a word is used once or twice, it’s not abbreviated.
Author Response
We appreciate your comments and suggestions regarding our manuscript “Molecular Biomarkers in Multiple Sclerosis and Its Related Disorders”. We’d like to apologize for the delay in revision, all the authors were extremely busy due to the COVID-19 circumstances. We have included a detailed summary of all the changes and modifications that were made to the best of our ability as per each of your recommendations. We hope you find our work to be noteworthy and acceptable for peer-review and eventual publishing.
Reviewer 2 Report
The manuscript ijms-825768 by Maryam et al report the molecular biomarkers in MS and other nervous system disorders.
Overall comments: Although the contains of the report are informative, the structure of the manuscript is not well organized.
Specific comments:
- The abstract consists of several words of general knowledge and simple list of the name of nervous system disorders which do not make any sense. Rewrite the whole section of Abstract.
- Only two section “introduction” and “discussion” exist in the manuscript which is very usual for scientific writing. The author might mix up the "introduction" and "results" section in this kind of writing.
- “1.3. Biomarkers in Treatment and Prognosis of MS” H3K9me2 is the general histone marker in all cells, cannot be any specific biomarker in diseases.
- “1.7. Primary Angiitis of the CNS “Recently the efficacy and safety of Mycophenolate Mofetil (MMF) for achieving and maintaining remission for childhood PACNS was proven [50].” The sentence does not make any sense.“Furthermore, high levels of neurofilament light chain (NFL) in the sera (CSF) indicates neuroaxonal damage, and its predictive value has been shown recently [51].” What did “neurofilament light chain (NFL)” refer to? protein name? or something else?
- “1.8. Polyarteritis Nodosa “Some of the suggested biomarkers in PAN are Erythrocyte Sedimentation Rate (ESR) and the matrix metalloproteinase-3 (MMP-3) which can both be used in the diagnosis, management and following up on the PAN prognosis” Erythrocyte Sedimentation Rate is a measurement of blood test, obviously not a molecular biomarkers.
- “1.9. Susac Syndrome” and “1.12. Arnold Chiari Malformation” The author did not mention any biomarker in the section. Why not delete section or fill up with new information?
- “1.11. Acute Disseminated Encephalomyelitis (ADEM) ““Several studies have shown that some new biomarkers like lower fractional anisotropy (FA), higher radial diffusivities (RDs), Myelin oligodendrocyte glycoprotein (MOG) antibody, serum IgG targeting myelin basic protein (MBP), and myelin-associated oligodendrocyte basic protein (MOBP) can be used to diagnose and differentiate ADEM from MS at the first attack [64, 66, 68, 69]”. Lower fractional anisotropy (FA), higher radial diffusivities (RDs)” is a scalar value, the author made the confusion with biomarker.
- “1.13. Neuromyelitis Optica (NMO) 1.14. Optic Neuritis” They seems different disease. Why their pathogenesis related to the same biomarker anti-MOG and anti-AQP4. The same situation happened in “1.17. Leber’s Hereditary Optic Neuropathy (LHON) 1.19. Metachromatic Leukodystrophy (MLD) “Why their pathogenesis related to the same biomarker Arylsulfatase A levels, and elevated CSF/urine sulfatide.
- “1.16. Cerebral Autosomal Dominant Arteriopathy with Subcortical Infarcts and Leukoencephalopathy (CADASIL)” Grammar mistake for the sentence.” Biomarkers showing promise are the Von Willebrand factor (vWF), Endothelial progenitor cells (EPCs), and Circulating progenitor cells (CPCs).”
Author Response

(The authors gave the same response as above.)

Reviewer 3 Report
ijms-825768; Gul ei al.
" Molecular Biomarkers in Multiple Sclerosis and Its Related Disorders: A Critical Review "
Multiple Sclerosis (MS) is a chronic autoimmune disease affecting the CNS which can lead to severe disability. In this review, the authors focus on pathophysiology, diagnosis, prognosis and therapeutic management approaches in patients with MS and its related disorders
My comments for the current version of the manuscript are as follows.
Minor
- I guess most of the readers (including me) may do not know that there are so many diseases related to MS. Therefore, it would be helpful if some backgrounds are first described in the section of Introduction.
- Recently, it was shown that T cells that target β-synuclein induced damage to the gray matter of the brain in MS (D. Lodygin et al. Nature 2019). Are there not any relevance to this result?
Author Response
Reviewers' Comments 3:
- “I guess most of the readers (including me) may do not know that there are so many diseases related to MS. Therefore, it would be helpful if some backgrounds are first described in the section of Introduction”.
Answer: Thank you for the suggestion, As highlighted in yellow on page 2 under the section “1. Multiple sclerosis (MS)”. We have added a more in-depth explanation about why there is such a wide range of diseases that may mimic MS. So as to provide the reader with some helpful background about the topic.
- “Recently, it was shown that T cells that target β-synuclein induced damage to the gray matter of the brain in MS (D. Lodygin et al. Nature 2019). Are there not any relevance to this result?”
Answer: Thank you for the great suggestion. As highlighted in yellow on page 2 under the section “1.1. Biomarkers in Pathophysiology of MS” and also in Table1; we have included a paragraph about the role of β-synuclein in MS pathology.
Round 2
Reviewer 2 Report
The author corrects with the errors which I raised in the manuscript. The quality of the manuscript had been improved than the previous version.
Author Response
thank you very much
Reviewer 3 Report
I agree that he authors responded to my previous comments.